# Effectiveness of traveller screening for emerging pathogens is shaped by epidemiology and natural history of infection

Katelyn M Gostic[1]*[†], Adam J Kucharski[2,3†], James O Lloyd-Smith[1,3]

[1]Department of Ecology and Evolutionary Biology, University of California, Los Angeles, Los Angeles, United States; [2]Centre for the Mathematical Modelling of Infectious Diseases, Department of Infectious Disease Epidemiology, London School of Tropical Hygiene and Medicine, London, United Kingdom; [3]Fogarty International Center, National Institutes of Health, Bethesda, United States

**Abstract** During outbreaks of high-consequence pathogens, airport screening programs have been deployed to curtail geographic spread of infection. The effectiveness of screening depends on several factors, including pathogen natural history and epidemiology, human behavior, and characteristics of the source epidemic. We developed a mathematical model to understand how these factors combine to influence screening outcomes. We analyzed screening programs for six emerging pathogens in the early and late stages of an epidemic. We show that the effectiveness of different screening tools depends strongly on pathogen natural history and epidemiological features, as well as human factors in implementation and compliance. For pathogens with longer incubation periods, exposure risk detection dominates in growing epidemics, while fever becomes a better target in stable or declining epidemics. For pathogens with short incubation, fever screening drives detection in any epidemic stage. However, even in the most optimistic scenario arrival screening will miss the majority of cases.

*For correspondence: kgostic@ucla.edu

†These authors contributed equally to this work

Competing interests: The authors declare that no competing interests exist.

## Introduction

International air travel drove the spread of SARS in 2003 and influenza A/H1N1p in 2009 (*Brockmann and Helbing, 2013*), and has since led to imported cases of influenza A/H7N9 (*William et al., 2015*), MERS-CoV (*Cauchemez et al., 2014*) and Ebola virus infection (*McCarthy, 2014*). Traveller screening policies, including fever screening and/or questionnaires at point of departure and/or arrival, have been proposed to limit the geographic spread of infection (*Malone et al., 2009*; *World Health Organization, 2009*; *Cowling et al., 2010*; *Khan et al., 2013*; *Bogoch et al., 2015*; *Centers for Disease Control and Prevention, 2014a*). Fever screening at point of arrival has been criticized, however, because long incubation periods and imperfect efficacy of fever screening devices reduce the probability of detecting symptoms in infected arriving passengers (*Pitman et al., 2005*; *Bitar et al., 2009*; *Mabey et al., 2014*). As the effectiveness of integrated screening programs will depend both on the pathogen-specific natural history of infection and epidemiological knowledge of exposure risk, as well as travel time and efficacy of screening methods, it is important to understand how these different factors contribute to screening effectiveness at departure and arrival.

During screening initiatives for influenza A/H1N1p, MERS-CoV, SARS-CoV and Ebola virus, large numbers of travellers were detained for in-depth assessment, but few or no cases were ultimately detected (*Table 1*). Although fever is the symptom most commonly measured during screening, it

**eLife digest** International air travel has contributed to the spread of several recent disease epidemics. For example, travelers infected with severe acute respiratory syndrome (or SARS) in 2003 carried the disease around globe. One infected air traveler can carry a disease to a new continent: in 2014, a man infected with Ebola in West Africa flew to the United States and infected two healthcare workers in Dallas during treatment.

Efforts to prevent the spread of SARS, Ebola and other disease outbreaks have included screening air passengers for infection prior to boarding, or immediately after arrival. In these situations, infrared thermometers are often used to check for symptoms of fever and passengers may be asked to fill out questionnaires to assess their risk of exposure to the disease.

However, the effectiveness of these airport screenings is questionable. 1000s of air travelers have been screened during several recent disease outbreaks, but few disease cases were detected. There are many reasons why an infected individual may be missed in airport screens. Passengers who have recently been infected may not yet display any symptoms and some passengers may be able to hide a fever or other symptoms by taking medication. Even if an individual has a fever, infrared thermometers will only detect it about 70% of the time. Also, screening questionnaires may miss passengers who are infected if they lie about any possible exposure to the disease.

Gostic et al. created a mathematical model to help assess how useful airport screening is for detecting cases of disease caused by the SARS coronavirus, Ebola, influenza H1N1 and several other viruses. The model reveals that the effectiveness of airport screening depends on several factors including: how long it takes for symptoms to develop after infection (the incubation period), how much is known about the virus and how it spreads, and whether the epidemic is still growing in size or is starting to slow down.

For influenza H1N1 and other viruses with short incubation periods, fever screening is the most successful method to detect cases throughout the epidemic. However, for viruses with long incubation periods—such as Ebola—questionnaires are more useful in the early stages of an epidemic when the number of cases is rapidly rising. Fever screening becomes more useful later in the epidemic when new cases start to fall because the people who are infected are more likely to be displaying symptoms.

Even so, Gostic et al. point out that in all of these scenarios airport screening will still miss many infected passengers. Thus, a challenge for future outbreaks will be to identify situations in which screening is worthwhile, and obtain better measurements of the factors that influence detection rates.

might not be detected in all infected individuals for several reasons. First, those with recent exposure may not yet have progressed to a symptomatic stage (*Pitman et al., 2005*; *Mabey et al., 2014*). Second, travellers might be symptomatic but not febrile; the probability a symptomatic patient will have a fever varies by pathogen (*Donnelly et al., 2004*; *Cao et al., 2009*; *Louie et al., 2009*; *Assiri et al., 2013*; *Cowling et al., 2013*; *Gao et al., 2013*; *Gong et al., 2014*; *Sun et al., 2014*; *WHO Ebola Response Team, 2014*). Third, the sensitivity of non-contact infrared thermometers (the devices most often used for airport fever screening) is limited, so passengers with fever may pass through symptom screening undetected (*Hausfater et al., 2008*; *Bitar et al., 2009*; *Nishiura and Kamiya, 2011*). Fourth, passengers may conceal fever and other symptoms during screening using antipyretic drugs (*Nishiura and Kamiya, 2011*). At the same time, fever is notoriously non-specific as a symptom, leading to high opportunity costs from detaining travellers with non-target illnesses (*Anderson et al., 2004*; *Gunaratnam et al., 2014*; *Mabey et al., 2014*).

Self-reporting of symptoms or potential recent exposure to infection via mandatory questionnaires is also a common component of traveller screening programs (*St John et al., 2005*; *Nishiura and Kamiya, 2011*; *Hale et al., 2012*; *Centers for Disease Control and Prevention, 2014a*; *Cho and Yoon, 2014*; *Gunaratnam et al., 2014*). Because information about risk factors does not depend on the presence of detectable symptoms at the time of screening, there is potential to identify a broader set of exposed travellers. However, epidemiological knowledge on factors linked to risk of infection is limited for some pathogens—particularly for novel emerging pathogens that are often the focus of

**Table 1.** Airport screening measures during past disease outbreaks

| Pathogen | Date | Location | Direction | Screened | Detained | Positive | Source |
|---|---|---|---|---|---|---|---|
| Influenza A/ H1N1p | 27 April–22 June 2009 | Auckland, New Zealand | Inbound | 456,518 | 406 | 4 | (*Hale et al., 2012*) |
| | 28 April–18 June 2009 | Sydney, Australia | Inbound | 625,147 | 5845 | 3 | (*Gunaratnam et al., 2014*) |
| | 28 April–18 June 2009 | Tokyo, Japan | Inbound | 471,733 | 805 | 15 | (*Nishiura and Kamiya, 2011*) |
| SARS Co-V | 5 April–16 June 2003 | Australia | Inbound | 1,840,000 | 794 | 0 | (*Samaan et al., 2004*) |
| | 31 March–31 May 2003 | Singapore | Inbound | 442,973 | 176 | 0 | (*Wilder-Smith et al., 2003*) |
| | 14 May–5 July 2003 | Toronto, Canada | Inbound | 349,754 | 1264 | 0 | (*St John et al., 2005*) |
| | 14 May–5 July 2003 | Toronto, Canada | Outbound | 495,492 | 411 | 0 | (*St John et al., 2005*) |
| MERS Co-V | 24 September 2012–15 October 2013 | England | Inbound | NR | 77 | 2 | (*Thomas et al., 2014*) |
| Ebola virus | August–September 2014 | Guinea, Liberia, Sierra Leone | Outbound | 36,000 | 77 | 0 | (*Centers for Disease Control and Prevention, 2014a*) |
| | 11 October–22 October 2014 | United States | Inbound | 762 | 3 | 0 | (*Apuzzo and Fernandez, 2014*; *CBS, 2014*) |

screening programs. Even for pathogens with well-characterized routes of transmission, not all cases will necessarily have a known source of exposure (*Lau et al., 2004*; *Cao et al., 2009*; *Tuite et al., 2010*; *Cowling et al., 2013*; *Gao et al., 2013*; *Gong et al., 2014*; *Sun et al., 2014*; *WHO Ebola Response Team, 2014*). Thus, the contribution of questionnaires to the overall effectiveness of traveller screening programs is unclear.

Screening initiatives have also been implemented both at points of departure and arrival. It has been suggested that departure screening is more efficient than entry screening because it needs to be implemented in only a few airports rather than globally (*Khan et al., 2013*; *Bogoch et al., 2015*), but there is often local political pressure for arrival screening as well. To understand how departure and arrival screening combine with pathogen natural history, epidemiological knowledge, efficacy of screening methodology, and human behavioral factors to determine overall screening outcomes, we developed a general modelling framework (*Figure 1*) for the screening process. We used this framework to assess outcomes for six pathogens of current or recent concern: influenza A/H7N9, influenza A/H1N1p, SARS-CoV, MERS-CoV, Ebola virus, and Marburg virus. By separating the contribution of different factors to the probability of detecting infectious travellers, we evaluated pathogen-specific strengths and weaknesses of different screening strategies. We considered scenarios in which the source epidemic is growing or stable, as epidemic phase influences the distribution of times since exposure in potential travellers. We also identified factors that could improve the effectiveness of screening programs for future emerging pathogens.

## Results

For each pathogen, the natural history of infection and state of epidemiological knowledge determined the potential for successful screening at various points in the process. For all six emerging pathogens considered here, the majority of identified cases exhibited a fever (*Figure 2A*). However, the proportion of confirmed cases who were aware of their exposure risk varied greatly. Influenza A/H1N1p,

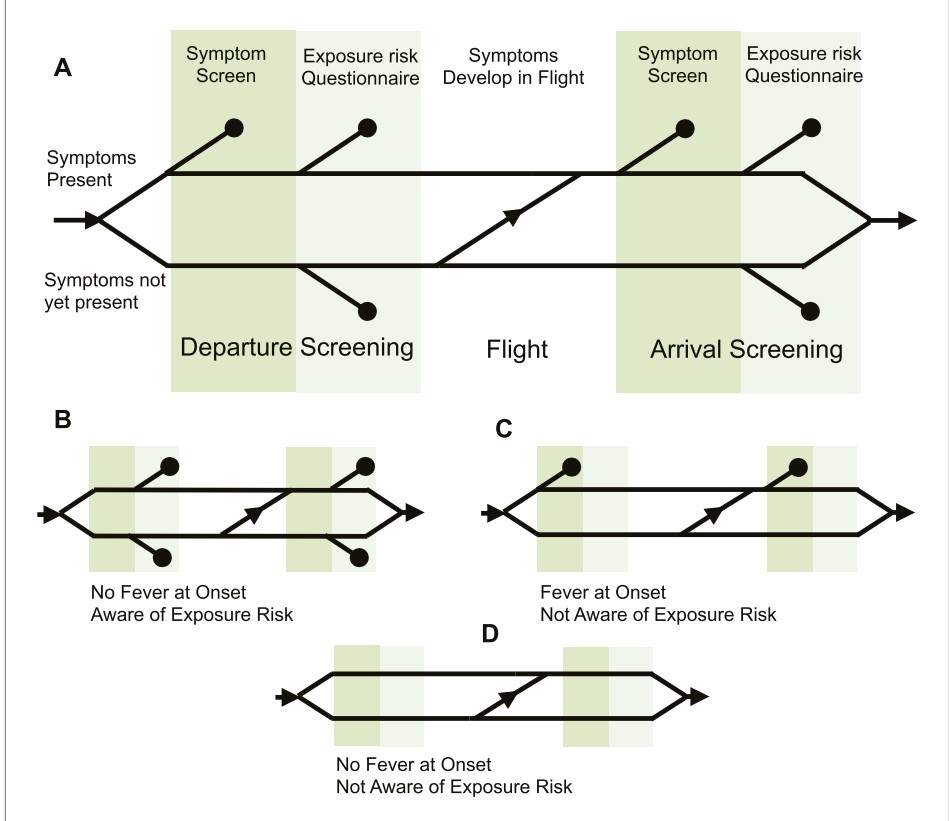

**Figure 1**. Model of traveller screening process. (**A**) Upon airport arrival, passengers passed through screening for fever, followed by screening for risk factors. We assumed a one-strike policy: passengers identified as potentially infected by any single screening test were detained. (**B**) Passengers who did not present with fever would always pass through symptom screening, but could still be identified during questionnaire screening. (**C**) Passengers who were not aware of exposure risk would always pass through questionnaire screening. (**D**) Passengers with neither fever nor knowledge of exposure would go undetected.

The following figure supplement is available for figure 1:

**Figure supplement 1**. Detailed model formulation with parameters.

which can have generic symptoms and can be transmitted via the airborne route, had the lowest reported proportion; Ebola virus, which requires close contact with infected individuals who have conspicuous symptoms, had the highest. Influenza A/H7N9, Marburg virus and SARS-CoV had similar proportions of cases that present with fever, and that had knowledge of exposure risk. We excluded MERS-CoV from the natural history space in *Figure 2A* because there are no established risk factors for exposure. Moreover, there was limited information available for the fever parameter for MERS-CoV: in a hospital outbreak of MERS-CoV, 20 out of 23 cases presented with fever at onset (*Assiri et al., 2013*); the small size of this sample means there is greater uncertainty surrounding the estimate for proportion of cases that exhibit fever.

The mean and variance of the incubation period have been recognized as key drivers of the effectiveness of fever screening at arrival, since shorter incubation periods mean a greater likelihood that travellers will progress to symptoms during travel (*Pitman et al., 2005*; *Al-Tawfiq et al., 2014*; *Mabey et al., 2014*). There was considerable variability in incubation period among different pathogens (*Figure 2B*). Influenza A/H7N9 and A/H1N1p have the shortest incubation periods, while Ebola virus and Marburg virus have the longest. For some pathogens, the estimated variance in incubation period could increase with the addition of more data, which would improve characterization of the tails of the distribution. For instance, the incubation period distribution for Marburg virus was estimated from just five cases with a single exposure opportunity (*Martini, 1973*); observing more cases might give rise to a right-skewed

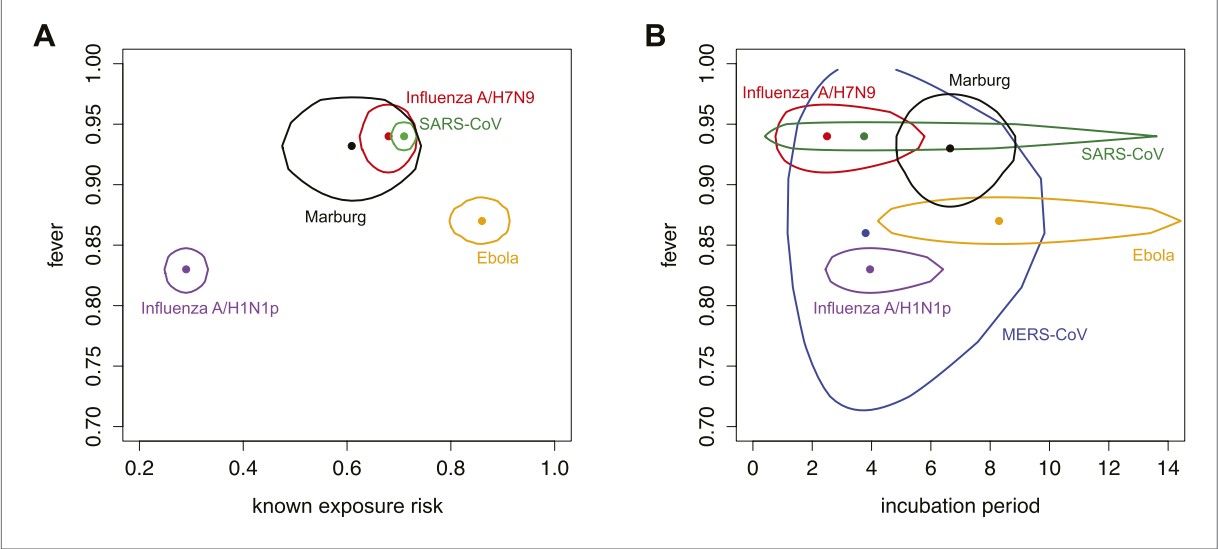

**Figure 2**. Parameters characterizing natural history of infection and epidemiological knowledge. (**A**) Proportion of infected individuals who report known exposure risk and show fever at onset. Point shows median estimate, using data in *Tables 2, 3*; circle shows joint 95% binomial confidence interval. Red, influenza A/H7N9; purple influenza A/H1N1p; blue, MERS; green, SARS; orange, Ebola; black, Marburg. (**B**) Incubation period and fever at onset. Point shows median estimate, circle shows joint 95% CI, generated using a binomial distribution for fever symptoms and fitted parametric distributions given by references in *Table 3* for incubation period.

distribution as seen for Ebola virus. Similarly, the incubation period distribution for MERS-CoV is determined using data from only 23 confirmed cases, and its variance might also expand with the addition of more data. The possibility of a lengthy incubation period presents challenges for symptom screening.

Our focus on the natural history and epidemiology of infection revealed the crucial influence of the time between exposure and the departing flight. When we included the natural history parameters in the model, (see 'Materials and methods') we found that the contributions of each component of a screening program depend strongly on the time between exposure to infection and intended departure from the airport (*Figure 3*). Individuals with more recent exposure were less likely to display symptoms at the time of screening, and hence less likely to be identified by fever screening at departure. For pathogens with long incubation periods, the marginal value of fever screening at arrival was also lower. Note, however, that the 70% efficacy of non-contact infrared thermoscanners means that arrival screening can contribute by catching symptomatic cases missed at departure. Thus, the bulk of the contribution of arrival fever screening is mediated by equipment efficacy rather than natural history.

Shortly after exposure, we found that detection was typically possible only by risk factor questionnaire screening, as most cases had not yet progressed to symptoms and were undetectable by fever screening (*Figure 3*). (Again, questionnaire screening at arrival contributes by catching some individuals who did not disclose their exposure risk at departure.) The duration of this phase depended on the incubation period, which is shortest for influenza A/H7N9 and A/H1N1p, and longest for Ebola and Marburg viruses; for MERS-CoV, despite a mid-length incubation period, questionnaire screening contributes nothing due to our ignorance of risk factors. As time since exposure elapsed, fever screening made a greater contribution to case detection, with pathogen natural history factors (i.e., incubation period, and fraction presenting with fever) becoming the primary determinants of screening effectiveness. We found similar qualitative patterns when we assumed reduced efficacy for fever screening devices (*Figure 3—figure supplement 1*), questionnaire reporting (*Figure 3—figure supplement 2*), or both tests (*Figure 3—figure supplement 3*).

The striking patterns in *Figure 3* highlight the important role of the 'infection age structure' (i.e., the distribution of times from exposure to departure) of the traveller population. As a basic consideration, cases are more likely to have progressed to severe disease or death as time since exposure increases, so the population of infected individuals able to attempt air travel will be skewed toward more recent exposures (i.e., younger infections). The distribution of time since exposure will be influenced by the epidemic phase in the source population (*Figure 4*, *Figure 4—figure supplement 1*).

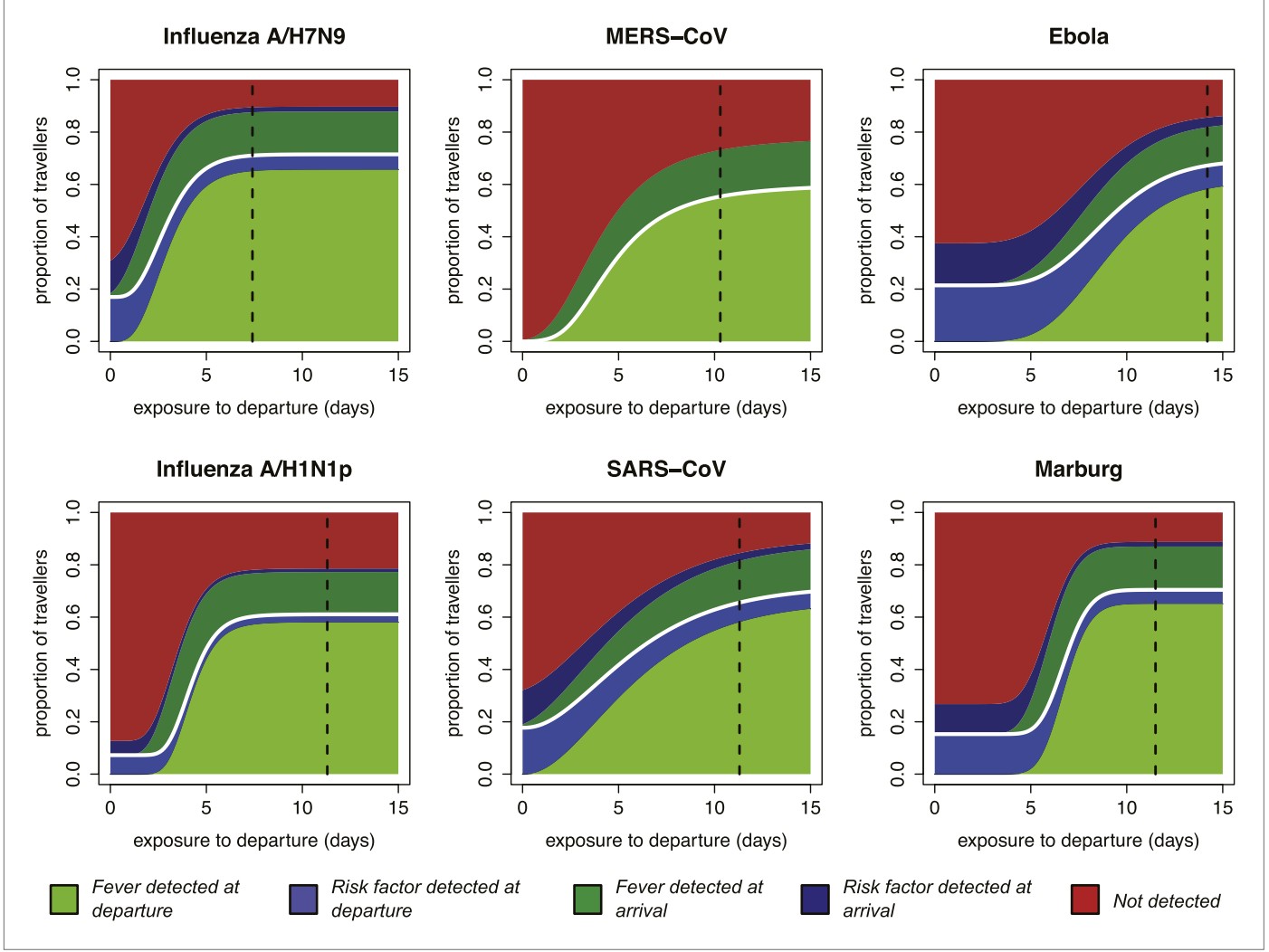

**Figure 3**. Impact of infection age on effectiveness of screening measures. Expected fraction of passengers detected by fever and risk factor screening, at arrival and departure, as a function of the time between an individual's exposure and the departure leg of their journey. We assume a 70% probability that fever screening will identify febrile patients, and a 25% probability that a traveller with a known history of risky exposure will report it on a questionnaire. We assume 24 hr travel time. The white lines denote the point at which travellers board their flight; the black dashed line shows the median time from exposure to hospitalization for each pathogen.

The following figure supplements are available for figure 3:

**Figure supplement 1**. Expected proportions detected by screening when efficacy of fever screening is 50% and proportion of cases with known exposure history who report correctly is 0.25.

**Figure supplement 2**. Expected proportions detected by screening when efficacy of fever screening is 70% and proportion of cases with known exposure history who report correctly is 0.1.

**Figure supplement 3**. Expected proportions detected by screening when efficacy of fever screening is 50% and proportion of cases with known exposure history who report correctly is 0.1.

Overall screening effectiveness was greater in stable than growing epidemics (*Figure 4A–B*). These gains were driven by increased potential for fever detection in stable epidemics, where cases are less likely to be recently exposed and asymptomatic. In contrast, exposure risk detection does not vary with epidemic phase because exposure risk awareness does not depend on the infection age distribution. Regardless of epidemic phase, the full screening program fails to detect at least 25% of infected

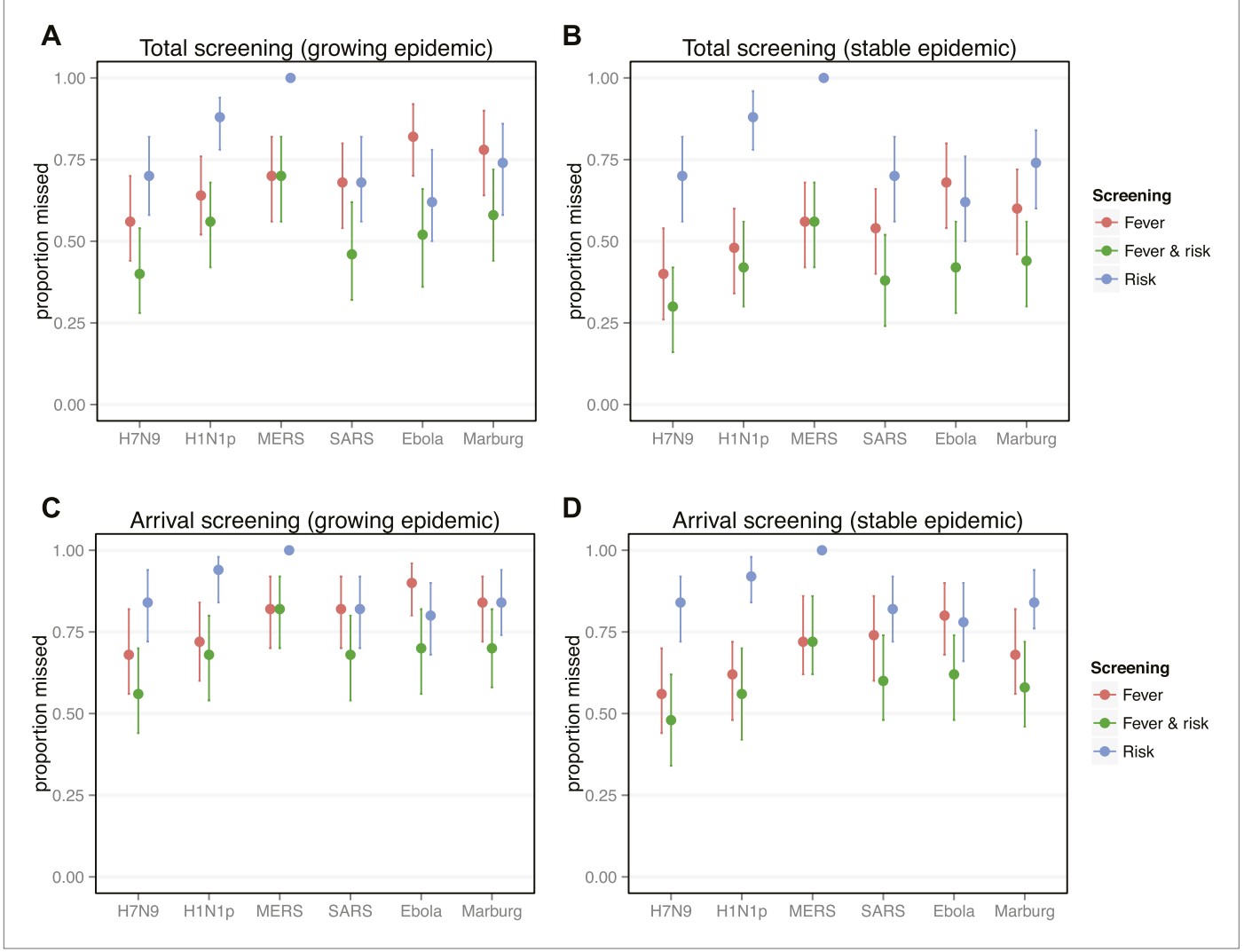

**Figure 4**. Proportion of infected travellers that would be missed by each of four screening scenarios. (**A**) Proportion of 50 infected travellers that would be missed by both departure and arrival screening in a growing epidemic. Figure shows three possible screening methods: fever screen, exposure risk questionnaire, or both. Lines show 95% bootstrapped CI. (**B**) Proportion of infected travellers missed by both departure and arrival screening in a stable epidemic. (**C**) Proportion of infected individuals who fly that are missed by arrival screening in a stable epidemic. (**D**) Proportion of infected arrivals missed by point of entry screening in a stable epidemic. We assume 25% probability traveller will report if they know exposure and 70% probability screening with identify visibly febrile patients. We assume R0 = 2 and a 24 hr travel time.

The following figure supplement is available for figure 4:

**Figure supplement 1**. Different time from exposure to departure functions used in model.

travellers, despite our optimistic assumptions. Focusing on the contribution made by screening at point of arrival, our projections suggest that arrival screening will still miss half to three-quarters of infected travellers that manage to complete their flights (*Figure 4C–D*). For pathogens with short incubation periods (i.e., influenza virus) fever detection was responsible for the majority of case identification in all epidemic phases. However, for pathogens with longer incubation periods (i.e., Ebola, Marburg, and SARS-CoV), exposure risk screening was responsible for half or more of case detection in growing epidemics. For these pathogens, fever detection was dominant only in stable epidemics (*Figure 4*).

## Discussion

We assessed the influence of pathogen natural history, knowledge of exposure risk, efficacy of screening techniques, and epidemic phase on the ability to detect infected passengers using integrated

traveller screening programs. By incorporating pathogen natural history and epidemiology into a mathematical model, we compared screening effectiveness for different pathogens, and showed that detection is driven by screening for exposure risk in travellers with recent exposure and by screening for fever in travellers with older infections. We found that natural history, epidemiological knowledge and epidemic phase combined to determine overall screening outcomes, as well as the relative contribution of each screening method to case detection. Exposure risk screening made a greater contribution to case detection during growing epidemics and for pathogens with longer incubation periods, and when exposure risk factors were well-characterized epidemiologically.

Our results highlight distinct taxonomic patterns in the effectiveness of screening measures. For influenza viruses, which have the shortest incubation periods, our findings suggest that fever screening would be responsible for the majority of case detection in any epidemic context. The picture is more subtle for filoviruses, which have longer incubation periods and distinctive symptoms. When the source epidemic is still growing and many travellers are recently exposed, our model suggests half or more of case detection for filoviruses will be driven by exposure risk screening; when the epidemic has stabilized, the shift in the age structure of infections means fever is likely to become the dominant mechanism of case detection (*Figure 4*).

Our analysis of coronaviruses illustrates two important features of the screening process. First, the variance of incubation period distributions can strongly modulate screening outcomes: the median incubation period for SARS-CoV is similar to influenza viruses, but because its distribution has a long right tail, the expected incubation time is longer. Thus, despite a short, influenza-like median incubation time, screening outcomes for SARS-CoV are filovirus-like and less favorable. Second, epidemiological knowledge is required to implement risk factor screening. Our results show that for pathogens with long incubation periods, early characterization of exposure risk factors is a powerful screening tool at the beginning of an outbreak; for these pathogens, robust characterization of and screening for specific exposure risk factors can contribute more to case detection than rapid implementation of fever screening (*Figures 3–4*). However, risk factor screening cannot improve screening effectiveness if factors that increase risk of exposure are not well characterized. For MERS-CoV, recent findings have strengthened the evidence that dromedary camels play a role in primary exposure of some cases, but it remains unclear whether the majority of transmission is driven by contact with infected humans or repeated zoonotic spillovers (*Azhar et al., 2014*; *Cauchemez et al., 2014*).

When novel pathogens emerge, it is important to prioritize case-control and other epidemiological studies to establish which factors contribute to exposure risk. As well as enhancing potential case detection using risk factor screening, such knowledge could have the additional benefit of promoting general awareness of exposure risk factors, and hence contribute to reductions in risky behavior in affected regions. Further, when risk factors are known there is potential to conduct extended follow-up with travellers who have exposure risks, but no symptoms at the time of flight. For example, patients with known risks for Ebola exposure should be monitored by local health authorities near their travel destination until the maximum plausible incubation period has elapsed (*Brown et al., 2014*).

Our results also illustrate that well-characterized exposure risks are not always sufficient: other factors of pathogen natural history can limit the potential effectiveness of risk detection. For example, exposure risks for influenza A/H7N9 are well defined and typically identifiable (exposure to live poultry or infected human contacts [*Cowling et al., 2013*; *Gao et al., 2013*]). However, because influenza A/H7N9 infection has a short incubation period and a high probability of generating febrile symptoms, fever screening becomes a potent tool within just a few days of exposure and eclipses the potential contribution of risk factor-driven detection. Thus, even during a growing epidemic, where the proportion of asymptomatic travellers is highest, fever detection is likely to remain the dominant mechanism of detection for pathogens with short incubation periods, such as the influenza viruses (*Figure 4A*).

We found that once the source epidemic stabilizes or begins to decline, screening has greater overall potential effectiveness: the infection age structure in a stable epidemic allows more fever-driven detection, which has much higher efficacy than risk screening (*Figure 4B*). This result reinforces other studies that emphasize the need to control outbreaks at the source during the growth phase (*Khan et al., 2013*; *Bogoch et al., 2015*; *Mabey et al., 2014*). Moreover, it suggests that stabilizing the growth of an epidemic could have the added benefit of making passenger screening more effective. Even in this stable phase, however, our results suggest that screening at point of arrival would still miss more than half of incoming infected passengers.

As well as being influenced by the natural history and epidemiological factors described above, the overall effectiveness of traveller screening depends on the efficacy of particular screening techniques. This efficacy in turn depends on a combination of instrument and human factors. We assumed that fever and risk factor screening are implemented with 70% and 25% efficacy, respectively, which we consider upper bounds (see 'Materials and methods'). Although our results are qualitatively insensitive to these assumptions (*Figure 3—figure supplements 1–3*), the quantitative results of our main analysis likely represent best-case scenarios. The estimated efficacy of fever screening reflects the sensitivity of non-contact infrared thermal scanner equipment, but human factors may further reduce the efficacy of screening techniques. For example, data from influenza A/H1N1p screening in Tokyo in 2009 suggested antipyretic drug use could have been widespread among febrile travellers (*Nishiura and Kamiya, 2011*). Additionally, outbreak-affected countries with heavily burdened public health systems may have limited resources to invest in departure screening, while limited preparation, awareness or focus may lower efficacy during arrival screening in countries outside the epidemic region.

Empirical evidence suggests that the majority of travellers with known exposure would not self-report (*Hale et al., 2012*; *Gunaratnam et al., 2014*), so the absolute effectiveness of risk factor screening in our model was lower than the effectiveness of symptom screening once cases progressed to onset, even if many exposed travellers were not yet symptomatic and symptom screening was only 70% effective (*Bitar et al., 2009*). For questionnaire-based screening, an essential unknown is the probability that travellers will divulge their exposure history if it puts them at risk of detainment or delay. We arrived at a rough, upper-bound estimate of 25% probability of honest reporting for the 2009 influenza A/H1N1p pandemic (see 'Materials and methods'), but we emphasize that this is a topic in need of further study. Even more valuable would be effective ways to motivate travellers to honestly report their risky exposures. Increasing honest exposure reporting not only has the potential to enhance detection of infected travellers, but is essential for implementation of follow-up monitoring of travellers who may have been exposed but have not yet developed symptoms.

Data from past and ongoing screening initiatives support our suggestion that outcomes predicted in this study should be interpreted as plausible best-case scenarios. For example, during the 2009 influenza A/H1N1 pandemic, arrival screening in Sydney, Australia detected 3 of an estimated 48 infected travellers, giving an empirical sensitivity of 7% (95% CI, 1–18%). This initiative used a combination of risk factor and fever detection in a growing epidemic, and yielded less favorable results than the effectiveness of 32% (95% CI 20–46%) predicted by our model. Also, questionnaire-based arrival screening for influenza A/H1N1 in Auckland, New Zealand detected 4 of 69 infected individuals, for a sensitivity of 6% (95% CI 2–14%). In this case, our model's predicted sensitivity (6%, 95% CI 2–16%) matches the observed pattern well.

Between August and January 2015, screening for Ebola in United States detected neither of two case importations and screening in the United Kingdom did not detect the single known case importation (*Department of Health, 2015*). Therefore observed sensitivity for Ebola screening has been 0%, which is lower than our model-predicted value of around 50%. While the comparison is not statistically significant with only three data points, these outcomes underscore that screening is inherently imperfect and can be expected to reduce—but not to prevent—disease importations.

These comparisons illustrate that actual screening efficacies and honest reporting fractions may vary considerably, and in some cases appear to be quite poor. Even under best-case scenario assumptions, our model suggests arrival screening will miss half or more of infected travellers. Thus, for screening to be implemented with reasonable effectiveness there is a need to identify behavioral incentives that encourage much better self-reporting and efficacy than the current data indicate.

There are some additional limitations to our framework. Because our model is structured so that fever screening precedes exposure risk screening, case detection through fever screening increasingly overlaps with potential detection through questionnaires as time since exposure increases. The model results appear to show that the effectiveness of exposure risk screening decreases with time since exposure (*Figure 3*), but in fact this shows that risk factor screening becomes increasingly redundant when passengers are subject to fever screening first. We have treated fever screening as the primary means of detection, as it is much easier to conclusively diagnose infection when symptoms are present (*Towner et al., 2004*; *Centers for Disease Control and Prevention, 2013*; *Centers for Disease Control and Prevention, 2014b*). The overall effectiveness of traveller screening, and the total proportion of cases detected before and after a flight, are independent of screening order. We did not consider the potential for case detection using symptoms other than fever. While other symptoms may aid

in case detection, many (coughing, sneezing, etc) are also non-specific and more easily concealed than fever in the early stages of infection (*Donnelly et al., 2004*; *Cao et al., 2009*; *Louie et al., 2009*; *Assiri et al., 2013*; *Gao et al., 2013*; *Gong et al., 2014*; *Sun et al., 2014*; *WHO Ebola Response Team, 2014*). Episodic symptoms such as vomiting, and internal symptoms such as gastrointestinal distress, would be difficult to detect via point-screening but could be incorporated into questionnaires.

We also concentrated on the sensitivity rather than specificity of screening measures. This is another respect in which our results should be considered a best-case scenario projection of detection outcomes, assuming that the financial and opportunity costs of imposing additional clinical assessment on a large number of uninfected individuals can be neglected. Past screening programs have been implemented at large financial cost and have delayed large numbers of travellers, while detecting only a few cases (*Table 1*), reflecting the fact that fever and many risk factors (e.g., contact with live poultry) have low positive predictive value for infection with rare pathogens. Though the high cost and low effectiveness of screening have been noted (*St John et al., 2005*; *World Health Organization, 2009*; *Cowling et al., 2010*; *Gunaratnam et al., 2014*; *Mabey et al., 2014*), to the best of our knowledge no formal cost analysis of traveller screening policies has ever been conducted. Such information would greatly aid future policy decisions about screening measures.

Screening at departure rather than arrival has been suggested as a more cost-effective and logistically feasible policy, as departure screening need be implemented only in affected regions, rather than globally (*Khan et al., 2013*; *Bogoch et al., 2015*). Our analysis suggests that arrival screening has the potential to make a non-negligible contribution to overall case detection, not only by detecting travellers who develop symptoms in flight, but also by detecting travellers who were missed by imperfect screening at departure. Hence, the additional benefit of arrival screening is greatest when efficacy of departure screening is relatively low, for example if potentially infected travellers primarily depart regions with limited public health re-sources and arrive in regions where public health resources are more abundant. Yet even costly policies that combine exit and arrival screening lack the potential to prevent all case importations. Our analysis suggests that in any context screening would miss a substantial proportion of infected travellers; this result is consistent with other analyses that highlight the limited effectiveness of screening (*Pitman et al., 2005*; *Bitar et al., 2009*; *Mabey et al., 2014*) and with previous or ongoing screening outcomes (*Table 1*). Policy makers should carefully consider whether resources are better spent on arrival screening, which will reduce but not eliminate the importation of cases, or instead on tracing and containing cases that inevitably do arrive.

Screening policies have been implemented during several recent epidemics (*Samaan et al., 2004*; *Pitman et al., 2005*; *St John et al., 2005*; *Nishiura and Kamiya, 2011*; *Hale et al., 2012*; *Khan et al., 2013*; *Centers for Disease Control and Prevention, 2014a*; *Gunaratnam et al., 2014*), and will likely continue to be discussed in response to future disease outbreaks. Certain aspects of screening, particularly fever screening at arrival, have been criticized as having little scientific justification (*Pitman et al., 2005*; *Bitar et al., 2009*; *Mabey et al., 2014*), but political leaders and health policy makers are likely to consider implementing screening programs when public pressure becomes intense. Thus there is a need to characterize the potential contributions of screening programs when implemented at different times, in different combinations, and for different pathogens; ultimately a quantitative understanding will be needed, to factor into cost-benefit calculations. In this study we begin to address these issues by demonstrating that screening outcomes depend strongly on pathogen natural history and epidemiological features, as well as human factors in implementation and compliance. Our results emphasize the need to characterize basic properties of emerging pathogens, as this knowledge can enhance disease control measures.

## Materials and methods

### Natural history and screening efficacy

Using previously published studies of influenza A/H7N9, influenza A/H1N1p, SARS-CoV, MERS-CoV, Ebola virus, and Marburg virus, we assembled a set of four parameters describing the natural history and epidemiology of each pathogen. To describe epidemiology we established the proportion of cases that had a known source of exposure, and for natural history we established the proportion of symptomatic cases that exhibited fever (*Table 2*). For pathogen natural history we also gathered estimates for incubation period (i.e., time from exposure to onset of symptoms), and the time from onset

**Table 2.** Natural history parameters: f is the proportion of cases with fever, g is the proportion of cases aware of exposure risk

| Pathogen | Parameter | Mean | Sample size | Reference |
|---|---|---|---|---|
| A/H7N9 | f | 0.79 | 85 | (*Cowling et al., 2013*) |
| | f | 1.00 | 46 | (*Gong et al., 2014*; *Sun et al., 2014*) |
| | f | 1.00 | 111 | (*Gao et al., 2013*) |
| | g | 0.75 | 123 | (*Cowling et al., 2013*) |
| | g | 0.56 | 111 | (*Gao et al., 2013*) |
| | g | 0.78 | 46 | (*Gong et al., 2014*; *Sun et al., 2014*) |
| A/H1N1 | f | 0.67 | 426 | (*Cao et al., 2009*) |
| | f | 0.89 | 1088 | (*Louie et al., 2009*) |
| | g | 0.29 | 426 | (*Cao et al., 2009*) |
| SARS | f | 0.94 | 1452 | (*Donnelly et al., 2004*) |
| | g | 0.29 | 1192 | (*Lau et al., 2004*) |
| MERS | f | 0.87 | 23 | (*Assiri et al., 2013*) |
| | g | 0 | 10,000 | (*Cauchemez et al., 2014*) |
| Ebola | f | 0.87 | 1151 | (*WHO Ebola Response Team, 2014*) |
| | g | 0.86 | 142 | (*Pattyn, 1978*) |
| Marburg | f | 0.93 | 129 | (*Bausch et al., 2006*) |
| | f | 0.47 | 15 | (*Bausch et al., 2003*) |
| | g | 0.67 | 39 | (*Roddy et al., 2010*) |

to hospitalization (*Table 3*), which we use to approximate the period after which most exposed individuals have progressed to severe illness and will not attempt travel (details in 'Appendix'). To estimate the proportion of individuals with known source of exposure for a particular pathogen, we identified the fraction of confirmed cases in descriptive epidemiological studies who reported contact with a known source of infection (e.g., poultry for influenza A/H7N9; close contact with an infected human for SARS-CoV, influenza A/H1N1p, Marburg and Ebola). As we could not find published estimates for the proportion of MERS-CoV cases with known source of exposure outside a hospital setting (*Assiri et al., 2013*), we assumed that exposure risk would not typically be known for MERS-CoV cases; this is consistent with recent publications highlighting the crucial knowledge gap in risk factors for MERS-CoV infection (*Al-Tawfiq et al., 2014*; *Zumla et al., 2014*).

A review of studies of non-contact infrared thermometer efficacy, when applied to forehead (as is typical for airport screening), suggested that the scanners had an average efficacy of 70% (*Bitar et al., 2009*). In our main analysis, we therefore assumed that the probability that febrile travellers would be detected by fever screening was 70%. This is an optimistic estimate, ignoring possible challenges in implementation in outbreak-affected regions and oversights made by device operators in arrival sites where risk may seem remote.

Another important parameter is the fraction of travellers who will report honestly about known exposure to risk factors in a screening questionnaire. This quantity is intrinsically difficult to measure, and to our knowledge it has not been estimated before. We estimate an upper bound on this quantity by drawing on information from studies of influenza A/H1N1p. As summarized in *Table 1*, one study from the early phase of the pandemic in China showed that 29% (95% binomial CI: 25–33%) of cases were aware of their exposure to earlier cases (*Cao et al., 2009*). Studies of influenza A/H1N1p screening in New Zealand (*Hale et al., 2012*) and Australia (*Gunaratnam et al., 2014*) estimated that self-reported exposure screening identified 3/45 and 4/69 infected passengers respectively. Assuming the lower limit of the CI, that is, 25% of the infected passengers knew about their exposure history, the New Zealand and Australia studies suggest that a proportion $3/(0.25 \times 45) = 0.27$ and $4/(0.25 \times 69) = 0.23$ of infected travellers who knew their exposure history reported so on the questionnaire. Based on this, in our main analysis we assumed a 25% probability of honest self-reporting

**Table 3.** Time from exposure to onset (i.e., incubation period) and onset to hospitalization for different pathogens

| Pathogen | Time from | Mean (days) | Reference |
|---|---|---|---|
| Influenza A/H7N9 | Exposure-to-onset | 4.3 | (*Cowling et al., 2013*) |
| | Onset-to-hospitalization | 5 | (*Gao et al., 2013*; *Sun et al., 2014*) |
| Influenza A/H1N1 | Exposure-to-onset | 4.3 | (*Tuite et al., 2010*) |
| | Exposure-to-onset | 2.05 | (*Ghani et al., 2009*) |
| | Onset-to-recovery | 7 | (*Tuite et al., 2010*) |
| SARS-CoV | Exposure-to-onset | 6.4 | (*Donnelly et al., 2003*) |
| | Onset-to-hospitalization | 4.85 | (*Donnelly et al., 2003*) |
| MERS-CoV | Exposure-to-onset | 5.2 | (*Assiri et al., 2013*) |
| | Exposure-to-onset | 5.5 | (*Cauchemez et al., 2014*) |
| | Onset-to-hospitalization | 5 | (*Assiri et al., 2013*) |
| Ebola | Exposure-to-onset | 9.1 | (*WHO Ebola Response Team, 2014*) |
| | Onset-to-hospitalization | 5 | (*WHO Ebola Response Team, 2014*) |
| Marburg | Exposure-to-onset | 6.8 | (*Martini, 1973*) |
| | Onset-to-hospitalization | 5* | |

*As there was limited data for onset-to-hospitalization for Marburg, we assumed the same value as for Ebola.

of exposure risk in each questionnaire. The assumption of independent decisions on each questionnaire is optimistic: it allows travellers who did not report honestly at departure to report honestly at arrival. The estimated probability is also optimistic, since we used a low-end estimate of the fraction of travellers who knew their exposure history: if more travellers were aware of their exposure, our estimate for the proportion who reported correctly would have been lower.

## Model

We used a probabilistic model to assess the influence of pathogen natural history and epidemiological factors on screening outcomes. Upon airport arrival, we assumed that passengers pass through screening for fever, followed by screening for risk factors (*Figure 1A*). We assumed a one-strike policy: infected passengers who were identified by any single screening test were successfully caught by the screening program. We used the incubation period distribution to estimate the proportion of passengers who progressed to symptom onset in flight.

The number of opportunities to detect each infected traveller varied depending on whether they displayed the symptom of fever and whether they knew their exposure history. We assumed passengers who did not present with fever would always pass through symptom screening, but could still be identified during questionnaire screening (*Figure 1B*). Passengers who are not aware of exposure risk will always pass through questionnaire screening (*Figure 1C*), and passengers with neither fever nor knowledge of exposure will go undetected (*Figure 1D*). The model is described in full in *Figure 1—figure supplement 1*. Source code for model analyses can be found in *Source Code 1*. This should be a citation for the source code file.

## Time from exposure to departure during growing epidemic and stable scenarios

Here we define the distribution of times from exposure to airport departure (i.e., the 'infection age distribution' for the traveller population). First, we consider a stable scenario, when the epidemic in the source population is neither growing nor shrinking. We assume that individuals are equally likely to depart at any point during the time period between exposure and hospitalization (when they would likely be too ill to fly) or death. Thus the infection age distribution of travellers will mirror that of the non-hospitalized case population. To model time to hospitalization we assume a simple case, in which all individuals progress to hospitalization after a fixed period of time; the probability density function for exposure-to-hospitalization is therefore represented by a delta function centered at the average

time from exposure to hospitalization in days derived from empirical studies, denoted D. Hence the time from exposure to departure has probability density function:

$$g(x) = \begin{cases} \dfrac{1}{D} & if\ 0 \leq x \leq D \\ 0 & else \end{cases}. \tag{1}$$

Next, we consider the distribution during the exponential growth phase of an epidemic. We assume the basic reproduction number, defined as the average number of secondary cases generated by each infectious case in the early period of the outbreak, is R0 and that the serial interval of the infection is D as above. The rate at which the number of infected individuals in the population changes is therefore given by:

$$\frac{dI}{dt} = \frac{R_0 I}{D}. \tag{2}$$

$$\Rightarrow I(t) = e^{\frac{R_0 t}{D}}. \tag{3}$$

If as before, we assume that no individual flies after time D, we have that the distribution of time from exposure to airport departure is:

$$f(x) = \begin{cases} \dfrac{1 - e^{\frac{R_0 t}{D}}}{1 - e^{R_0}} & if\ 0 \leq x \leq D \\ 0 & else \end{cases}. \tag{4}$$

## Detailed model formulation

The model considers a population of travellers, and only considers those travellers that are infected with the pathogen of interest. The probability density functions derived above are used to describe the distribution of times since exposure of individuals attempting travel, which we denote $\theta(d)$. Individuals of a given infection age are classified as symptomatic or asymptomatic at the time of intended departure using the cumulative distribution function of the incubation period distribution, denoted $\delta(d)$.

Following symptom classification, travellers are assigned to one of four detectability classes: fever present with symptom onset, aware of risk factors, neither or both. The proportion of individuals assigned to any detectability class is assumed independent of time since exposure. We also assume that presence of fever is independent of exposure risk awareness, so that the fraction of travellers in each respective category is: $f(1-g)$; $(1-f)g$; $(1-f)(1-g)$; $fg$. See *Table 2* and *Figure 2* for parameter estimates.

Travellers subsequently pass through fever screening, followed by risk factor screening, but are not affected by screening phases that are incompatible with their detectability class. For each phase of the screening process, the probability of detecting a case (given that the case has fever or risk factor, and hence could be detected by that screening modality) is modulated by an efficacy parameter, $\varepsilon$. The efficacy parameters have two subscripts: the first subscript is f, denoting fever screening, or g, denoting exposure risk screening; the second subscript is d, denoting departure screening, or a, denoting arrival screening.

After departure screening for symptoms and risk factors, cleared passengers board the flight and some progress to symptoms during travel. Passengers who were symptomatic at departure but were not detected by screening remain symptomatic with probability 1. Passengers who were not symptomatic at departure develop symptoms in flight with probability S:

$$S = \frac{\delta(d) - \delta(d + \Delta d)}{1 - \delta(d)}. \tag{5}$$

Finally, passengers pass through arrival screening using the same framework described above, using the appropriate efficacy parameters.

To determine the probability that an individual with a known time since exposure is detected at arrival or departure by fever screening or risk detection, we ran the model using single, fixed times since exposure. Then, to analyze outcomes within a population of travellers with mixed times since exposure, we used the infection age distributions described in *Equations 1 and 4* to estimate overall detection effectiveness. As noted in the main text, during past screening initiatives the absolute number of infected travellers was very low: for pandemic influenza A/H1N1 only 45 and 69 cases were imported in to Sydney and Auckland, respectively (*Hale et al., 2012*; *Gunaratnam et al., 2014*). Between August and November 2014, only two Ebola importations occurred, both in the United States (*Brown et al., 2014*). Therefore we chose to simulate detection for a population of 50 infected individuals, and recorded the number of individuals identified by screening. We then repeated the simulation 2000 times and reported the median case detection fraction and 95% confidence interval of the proportion of identified cases across all simulations.

## Acknowledgements

We thank the Lloyd-Smith lab for valuable comments on this work.

## Additional information

### Funding

| Funder | Grant reference number | Author |
| --- | --- | --- |
| National Institutes of Health | T32-GM008185 | Katelyn M Gostic |
| Medical Research Council | MR/K021524/1 | Adam J Kucharski |
| U.S. Department of Homeland Security | RAPIDD Program | James O Lloyd-Smith, Adam J Kucharski |
| Fogarty International Center | RAPIDD Program | James O Lloyd-Smith, Adam J Kucharski |
| National Science Foundation | EF-0928690 | James O Lloyd-Smith |

The funders had no role in study design, data collection and interpretation, or the decision to submit the work for publication.

### Author contributions

KMG, AJK, Conception and design, Acquisition of data, Analysis and interpretation of data, Drafting or revising the article; JOL-S, Conception and design, Analysis and interpretation of data, Drafting or revising the article

## Additional files

### Supplementary file

• Source code 1. (**A**) Internal Functions Filename—Code_distribution_functions.R. This script contains user-defined functions and distributions that are called by the master script. (**B**) Screening Model Filename—Code_Screening_model.R. This script defines the core probabilistic model described in this manuscript. This function is called by the master script. (**C**) Master Script Filename—Plot_results.R. This script integrates all the provided code to perform analyses and generate figures presented in this manuscript.

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
