## [Decision Letter]

Thank you for sending your work entitled “Effectiveness of traveller screening for emerging pathogens is shaped by epidemiology and natural history of infection” for consideration at *eLife*. Your article has been favorably evaluated by Prabhat Jha (Senior editor) and 3 reviewers, one of whom is a member of our Board of Reviewing Editors.

The following individuals responsible for the peer review of your submission have agreed to reveal their identity: Simon Hay (Reviewing editor and peer reviewer) and Nick Golding (peer reviewer). One other reviewer remains anonymous.

The Reviewing editor and the other reviewers discussed their comments before we reached this decision, and the Reviewing editor has assembled the following comments to help you prepare a revised submission.

The revisions requested are relatively trivial: enhanced discussion points from reviewers 1 and 2. In addition, reviewer 3 requires some additional simulations to be done which we agree would be prudent.

*Reviewer #1*:

I thought this was a well-conceived, written and executed piece of work. It was nuanced correctly with many of the important caveats adequately covered in the discussion. I have little to criticize.

I would perhaps seek a few points of elaboration in the Discussion.

First: Many studies including (doi: 10.1016/S0140-6736(1461828-6)), of which I have been part of, have recommend exit screening as a logistically more viable option due to potentially infected people being clustered at the “epidemic” source. Some discussion of this in relation to these findings would be interesting and indeed a bit more reference to the ongoing screening efforts in the current outbreak.

Two: We now have quite a lot of experience of screening (8–10 months and known airport import events). Do these search efforts and number of detected cases, chime with your findings?

Three: As alluded to in the Discussion, it is often a political necessity to screen, regardless of the likely efficiency. Has anyone ever done a costing study of this? I am not proposing that you do one but perhaps discuss the need for this in the context of your results.

*Reviewer #2*:

This manuscript reports a modelling analysis estimating the effectiveness of airline screening for infectious diseases and the impact of disease-specific characteristics on this efficacy. The analysis considers 6 diseases of major global concern, including Ebola and MERS-CoV, which are currently undergoing outbreaks of significant global concern. This is obviously a very timely topic and the questions posed and answered in this paper will be of great interest to the many public health officials attempting to find empirical data to guide screening policies, as well as to other researchers. The methodology employed is sound, the results are clearly presented and adequately discussed and the manuscript is very well written throughout. I therefore recommend publication of the article in its current form.

The authors may also wish to consider sharing the source code used to fit the models in order to help public health researchers and policy makers to update these estimates as more data become available and as new diseases arise.

*Reviewer #3*:

The authors propose a simple yet interesting analysis of the level of detection from airport screening for diseases with different epidemiological history. They apply their method to estimate infected passenger detection via temperature check (for fever) and risk of exposure questionnaire for recent pandemic strains of flu, SARS, MERS-CoV, Ebola, and Marburg virus. The authors use parameters to characterize each of those diseases from the literature, clearly indicating the cases where the current knowledge is not sufficient to have a reasonable confidence interval for those parameters.

The diseases studied have different characteristic periods between exposure and symptoms onset, probability to develop fever, and public awareness of risk of exposure (i.e., individual knowledge of having been or not potentially exposed to the pathogen). This variability allows for the study of screening methods efficacy in different and realistic scenarios.

The authors also provide sensitivity analysis on the efficacy of fever screening equipment and proportion of cases with known exposure history who report correctly.

Their results are consistent with the scientific community assumption that such measure is strongly effective only when the disease has a short incubation period or, equivalently, when passengers travel only relatively long after being exposed. Such result is important for policy makers so that they take informed action. Nonetheless, the authors should emphasize the importance of questionnaires for follow up measures such as establishing health monitoring. That is, even if not for actual infected case detection at the port of entry, but for detection of possible exposure.

On the other hand, the broad tail on the distributions of fever occurrence and/or incubation period, combined with the other probabilities involved in the study, calls for a large number of trials in order to draw confident conclusions. The authors report the usage of 50 infected travelers for each scenario, for each disease, and they do not mention whether this process is repeated a certain number of times or not. The authors mention the use of bootstrapping to estimate the confidence intervals, but never mention the number of simulations used. One might argue that for some of the diseases studied the expected number of infected international passengers is small, thus justifying the relatively small number of infected travelers used in those particular cases. Nonetheless, in each scenario and for each disease, the process should be repeated a large number of times and have the results drawn from the statistics thereof, especially since the article goal is to provide a general picture of screening efficacy in future events. The authors should rerun their simulations taking that into account or explicitly mention the number of trials and number of infected passengers used. Hopefully it won't affect the general picture provided, but it should be done for consistency.

---

## [Author Response]

We have revised the manuscript according to the comments from all reviewers. In particular, a comment from reviewer 3 pointed out that our initial bootstrapping strategy for the results presented in Figure 4 did not fully capture both parameter-driven and sampling variability. We repeated these simulations with appropriate modifications. This modification has led to slightly larger confidence intervals for some screening scenarios, but our overall conclusions remain unchanged. Additionally, reviewer 1 suggested several points for addition to the Discussion. We agreed that each of these would enhance our analysis and have made appropriate changes to the text. Finally, reviewer 2 suggested we publish our source code as a supplement to this manuscript. We agree that making our source code available would be valuable and have included three commented scripts as a supplement to our revised submission.

Reviewer #1:

*[…] I would perhaps seek a few points of elaboration in the Discussion*.

*1) First: Many studies including (*doi: 10.1016/S0140-6736(1461828-6)*), of which I have been part of, have recommend exit screening as a logistically more viable option due to potentially infected people being clustered at the “epidemic” source. Some discussion of this in relation to these findings would be interesting and indeed a bit more reference to the ongoing screening efforts in the current outbreak*.

We agree that this is an important point and have added text discussing this pertinent, but complex issue (shown below). Briefly, our analysis highlights the cost-effectiveness trade-off that underlies the decision to rely on exit screening alone or to implement arrival screening in addition to exit screening. A formal analysis of this trade-off is beyond the scope of this study, but would be a useful tool in policy decisions.

“Screening at departure rather than arrival has been suggested as a more cost-effective and logistically feasible policy, as departure screening need be implemented only in affected regions, rather than globally (Bogoch et al., 2012; [30]). On the other hand, our analysis suggests that arrival screening has the potential to make a non-negligible contribution to overall case detection, potentially detecting an additional 20% of infected travellers. Notably, arrival screening can contribute not only by detecting travellers who develop symptoms in flight, but also by detecting travellers who were missed by imperfect screening at departure. Hence, the additional benefit of arrival screening is greatest when efficacy of departure screening is relatively low, for example if potentially infected travellers primarily depart regions with limited public health resources and arrive in regions where public health resources are more abundant. Yet even costly policies that combine exit and arrival screening will not prevent all case importations. Our analysis suggests that even under optimistic assumptions screening at both departure and arrival would miss a quarter or more of infected travellers: this result is consistent with other analyses that highlight the limited effectiveness of screening (40; 8; 33) and with outcomes of past or ongoing screening programs (see Table 1). Because the marginal cost of additional case detection with arrival screening is likely to be high, policy makers should carefully consider whether resources are better spent on arrival screening, which will reduce but not eliminate the importation of cases, or instead on tracing and containing cases that inevitably do arrive.”

*2) Two: We now have quite a lot of experience of screening (8–10 months and known airport import events)*. *Do these search efforts and number of detected cases, chime with your findings?*

Again, we agree that this discussion point would enhance our paper. However after reviewing the literature and press reports on ongoing Ebola screening initiatives we found that only three infected individuals have passed through airport screening: two were destined for the United States and one was destined for England. Each of these travellers passed through both departure and arrival screening undetected (11). We have included this observation, but because the sample size for Ebola importations is so small, we used data from previous influenza A/H1N1 screening initiatives for comparison with model-predicted outcomes. We have added the text below in reference to this comparison:

“Data from past and ongoing screening initiatives support our suggestion that outcomes predicted in this study should be interpreted as plausible best-case scenarios. For example, during the 2009 influenza A/H1N1 pandemic, arrival screening in Sydney, Australia detected 3 of a possible 48 infected travellers, giving an empirical sensitivity of 6.7% (95% CI, 1.4–18.3%). […] Thus, for screening to be implemented with reasonable effectiveness there is a need to identify behavioral incentives that encourage much better self-reporting and fever screening efficacy than the current data indicate.”

*3) Three: As alluded to in the Discussion, it is often a political necessity to screen, regardless of the likely efficiency. Has anyone ever done a costing study of this? I am not proposing that you do one but perhaps discuss the need for this in the context of your results*.

We have addressed this issue by adding the following to our Discussion:

“Though the high cost and low effectiveness of screening have been noted (43; 51; 19; 27; 33), to the best of our knowledge no formal cost analysis of traveller screening policies has ever been conducted. Such information would greatly aid future policy decisions about screening measures.”

Reviewer #2:

*This manuscript reports a modelling analysis estimating the effectiveness of airline screening for infectious diseases and the impact of disease-specific characteristics on this efficacy. The analysis considers 6 diseases of major global concern, including Ebola and MERS-CoV, which are currently undergoing outbreaks of significant global concern. This is obviously a very timely topic and the questions posed and answered in this paper will be of great interest to the many public health officials attempting to find empirical data to guide screening policies, as well as to other researchers. The methodology employed is sound, the results are clearly presented and adequately discussed and the manuscript is very well written throughout. I therefore recommend publication of the article in its current form*.

*The authors may also wish to consider sharing the source code used to fit the models in order to help public health researchers and policy makers to update these estimates as more data become available and as new diseases arise*.

We agree that sharing this source code would be beneficial and have published the code as a supplement to this manuscript.

Reviewer #3:

*[…] Their results are consistent with the scientific community assumption that such measure is strongly effective only when the disease has a short incubation period or, equivalently, when passengers travel only relatively long after being exposed. Such result is important for policy makers so that they take informed action*.

*1) Nonetheless, the authors should emphasize the importance of questionnaires for follow up measures such as establishing health monitoring. That is, even if not for actual infected case detection at the port of entry, but for detection of possible exposure*.

This is a valuable suggestion and we have added the following discussion point to the text:

“Further, when risk factors are known there is potential to conduct extended follow-up with travellers who have exposure risks, but no symptoms at the time of flight. For example, according to current US guidelines, patients with known risks for Ebola exposure should be monitored by local health authorities near their travel destination until the maximum plausible incubation period has elapsed (11).”

*2) On the other hand, the broad tail on the distributions of fever occurrence and/or incubation period, combined with the other probabilities involved in the study, calls for a large number of trials in order to draw confident conclusions. The authors report the usage of 50 infected travelers for each scenario, for each disease, and they do not mention whether this process is repeated a certain number of times or not. The authors mention the use of bootstrapping to estimate the confidence intervals, but never mention the number of simulations used. One might argue that for some of the diseases studied the expected number of infected international passengers is small, thus justifying the relatively small number of infected travelers used in those particular cases. Nonetheless, in each scenario and for each disease, the process should be repeated a large number of times and have the results drawn from the statistics thereof, especially since the article goal is to provide a general picture of screening efficacy in future events. The authors should rerun their simulations taking that into account or explicitly mention the number of trials and number of infected passengers used. Hopefully it won't affect the general picture provided, but it should be done for consistency*.

Thank you for raising this point. Our confidence intervals in the original manuscript reflected uncertainty in the biological parameters used in the model, but not the uncertainty generated by repeatedly sampling 50 passengers. We have therefore repeated our analysis to incorporate both sources of uncertainty. To generate the results presented in the revised Figure 4, we therefore simulated 2000 trips made by a population of 50 infected travellers, which should sufficiently capture the distribution of screening outcomes. We have added text to describe this method to Appendix 2. The overall conclusions remain the same, but the confidence intervals are now slightly broader for some scenarios.

We chose to model 50 infected travellers in this study precisely because this number is consistent with the number of infected air travellers during past influenza A/H1N1 screening initiatives. As noted in the text, an estimated 69 infected travellers passed through Auckland, New Zealand, and 48 through Sydney, Australia during the A/H1N1 outbreak. During ongoing Ebola screening initiatives in the United States and the United Kingdom, only three infected travellers have passed through arrival screening. It is possible that the number of Ebola cases detained by departure screening in affected West African countries is higher, though currently available data does not suggest that large numbers of infected travellers have been detained. According to the most recent statistics we were able to find, as of October 8, 2014 no travellers with confirmed Ebola infection had been detained by exit screening in affected West African countries (CDC, 2014). The release of more current data would aid assessments of screening effectiveness at arrival or departure.